

**The non-conservative distribution pattern of organic matter in**
**Rajang, a tropical river with peatland in its estuary**
Zhuoyi Zhu[1]*, Joanne Oakes[2], Bradley Eyre[2], Youyou Hao[1], Edwin Sien Aun Sia[3], Shan Jiang[1],
Moritz Müller[3], Jing Zhang[1]
1. State Key Laboratory of Estuarine and Coastal Research, School of Oceanography, East China
Normal University, Shanghai, 200241, China
2. Centre for Coastal Biogeochemistry, School of Environment, Science and Engineering, Southern
Cross University, Lismore NSW, 2480, Australia
3. Swinburne University of Technology, Faculty of Engineering, Computing and Science, Jalan
Simpang Tiga, Kuching, 93350, Sarawak, Malaysia
*corresponding author: Z.Y. Zhu, zyzhu@sklec.ecnu.edu.cn; zhu.zhuoyi@163.com
**Abstract**
South-east Asian peatland-draining rivers have attracted much attention due to their high
dissolved organic carbon (DOC) yield and high $CO_2$ emissions under anthropogenic activities. In
August 2016, we carried out a field investigation of the Rajang river and estuary, a tropical system
located in Sarawak, Malaysia. The Rajang has peatland in its estuary while the river basin is covered
by tropical rainforest. DOC $\delta^{13}$C in the Rajang ranged from -28.7‰ to -20.1‰ and a U-shaped trend
from river to estuary was identified. For particulate organic carbon (POC), the $\delta^{13}$C ranged between
−29.4‰ to −31.1‰ in the river and a clear increasing trend towards more $\delta^{13}$C -enriched with higher
salinity existed in the estuary. In the estuary, there was a linear conservative dilution pattern for
dissolved organic matter composition (as quantified by D/L amino acids enantiomers) plotted
against DOC $\delta^{13}$C, whereas when plotted against salinity dissolved D/L amino acids enantiomers
values were higher than the theoretical dilution value. Together, these data indicate that the addition
of DOC in estuary (by peatland) not only increased the DOC concentration, but also altered its
composition, by adding more bio-degraded, $^{13}$C-depleted organic matter into the bulk dissolved
organic matter. Alteration of organic matter composition (adding of more degraded subpart) was
also apparent for the particulate phase, but patterns were less clear. The Rajang was characterized
by DOC/DON ratios of 50 in the river section, with loss of DON in the estuary increased the ratio
to 140, suggesting the unbalanced export pattern for organic carbon and nitrogen, respectively.
Under anthropogenic activities, further assessment of organic carbon to nitrogen ratio is needed.
**Keyword**
Amino acids enantiomers, DOC, POC, stable carbon isotope, Rajang, peatland





## 1. Introduction


Fluxes and cycling of organic matter (OM) in rivers and estuaries are important influences on
global biogeochemical cycles and climate change. In river basins, vascular plants are the ultimate
sources of organic matter (Hedges and Man, 1979), but algae, moss and bacteria are also important
(Hernes et al., 2007). As well as providing a source of OM, bacteria may also strongly modify the
composition of organic matter within a river and its resistance to degradation. The lability of organic
matter determines how rapidly organic carbon will be transformed into inorganic carbon ($CO_2$),
which can vary from hours to millions of years. The lability of organic matter therefore plays a role
in determining whether organic matter is either a source or a sink of carbon in the atmosphere (Zhang
et al., 2018). Based on $^{14}C$ of organic carbon, Mayorga et al. (2005) determined that the degradation
of recently synthesized organic matter in the river basin was the main reason Amazonian river
waters were supersaturated in $CO_2$, and hence the a source of atmospheric $CO_2$. This highlights the
potential importance of organic matter stability for carbon cycling within river systems. Nitrogen is
another important element in organic matter, which is not independent from carbon, but instead is
closely combined with carbon in various chemical compounds (like amino acids). Due to the nature
of these specific compounds, the behavior of bulk carbon and nitrogen can differ substantially. In
basins with peatland, the leaching of DOC is related to the status of peatland (disturbed vs
undisturbed), whereas the leaching of dissolved organic nitrogen (DON) is controlled by the soil
inorganic nitrogen content (Kalbitz and Geyer, 2002). The different leaching mechanisms of organic
carbon and nitrogen indicates that the comparison of these two elements would deepen our
understanding of organic matter cycles.
Tropical south-east Asian rivers play an important role in both dissolved and particulate organic





matter export (Baum et al., 2007; Huang et al., 2017; Müller et al., 2016). Located in Sarawak,
Malaysia (Fig. 1a), the turbid Rajang river (hereafter refer to as the Rajang) is the longest river in
Malaysia. The Rajang flows through tropical rainforest, and peatland and mangroves are distributed
in the estuary (downstream of Sibu; Fig. 1b). A dam was constructed in the upper reaches of the
Rajang in 2015, but the total suspended matter (TSM) in the river downstream of Kapit remains at
100 – 200 mg/L in recent years (Müller-Dum et al., 2019). Dilution of terrestrial organic matter in
the adjacent coast is expected, while turbid river water strongly limits apparent organic matter
photo-degradation within the river and estuary, leaving the stage of fluvial organic matter alteration
to bacteria utilization and abiotic process like desorption/adsorption between particulate and
dissolved phase (Martin et al., 2018). Further, dissolved oxygen is negatively related to $pCO_2$, likely
due to in-stream heterotrophic respiration (Müller-Dum et al., 2019). In the Rajang brackish estuary,
where peatland is located, addition of peatland DOC into river water is suggested by the non-
conservative mixing pattern of DOC with increasing salinity (Martin et al., 2018), whereas removal
of DON in the Rajang estuary is suggested by nitrogen stable isotopes (Jiang et al., 2019).

While stable isotopes of carbon and nitrogen are useful tools for tracing organic matter, amino

acids (AAs) are the most important organic carbon and nitrogen carriers that have been chemically
identified, accounting for up to ~100% of the particulate nitrogen in aquatic environments, and up
to nearly half of the particulate organic carbon pool (Jennerjahn et al., 2004). Due to the selective
removal and accumulation of certain amino acids, amino acids are important biomarkers in early
diagenesis, allowing quantification of organic matter lability/resistance (Dauwe and Middelburg,
1998; Kaiser and Benner, 2009). With the exception of glycine, amino acids are chiral. L forms of
amino acids are from animals, plants and plankton, whereas D forms mainly come from bacteria,



and are key chemical compounds in peptidoglycan, which forms the basic structure of bacterial cell
membranes (Vollmer et al., 2008). Due to the key role of bacteria in OM alteration and early
diagenesis, D-AAs (D forms AAs) tend to accumulate during OM degradation. A higher ratio of D-
to L-AAs (D/L ratio) therefore indicates more that OM is more refractory (Davis et al., 2009). As a
non-protein amino acid, accumulation of GABA (γ─aminobutyric acid) is also highly related to
OM degradation (Davis et al., 2009). Conversely, a lower D/L ratio and GABA% indicates that OM
is relatively less degraded, and hence more labile. In river waters, elevated D-AAs also indicates
the presence of soil humic substances, which is a product of bacteria and their detritus (Kimber et
al., 1990).
Tropical rivers are dominated by refractory (or bio-degraded) organic matter, yet labile OM is
also known to play a role in river carbon cycles (Mayorga et al., 2005). It is hence expected that the
fluvial organic matter in the river would be a mixture of labile organic matter (that can be respired
to support $pCO_2$) and refractory terrestrial organic matter (that will be diluted/degraded after
entering the sea (Martin et al., 2018), while in the estuary there would be addition of dissolved OM
from peatland/mangrove (Dittmar et al., 2001b; Müller et al., 2016). Previous studies of OM in
south-east Asian rivers mainly focused on its bulk concentrations, ages, or optical properties (Martin
et al., 2018 and ref. therein). The use of biomarker approaches has been very limited (Baum et al.,
2007; Gandois et al., 2014). Given the processes described above and their potential contribution to
the carbon (Müller-Dum et al., 2019) and nitrogen cycles (Jiang et al., 2019), it is somewhat
surprising that there has been limited application of amino acid approaches, including D-AAs, to
investigate organic matter composition and the role of estuarine peatland/mangrove in OM
regulation (Jennerjahn et al., 2004). South-east Asian rivers are subject to multiple stressors due to





increasing anthropogenic activities in both their riverine (e.g., damming, logging/secondary
plantation) and estuarine sections (e.g., drainage, and oil palm plantations) (Hooijer et al., 2015).
AAs enantiomers and carbon/nitrogen isotopes have the ability to provide molecular level evidence
for the impact of these stressors on carbon and nitrogen cycling and bulk biogeochemistry, as well
as insight into the mechanisms underlying such changes.
In this study, we carried out a field investigation in the Rajang in August 2016, from Kapit to
S1 station, located on the coast of the South China Sea adjacent to the Rajang (Fig. 1b). AAs
enantiomers and $\delta^{13}C$ of DOC were used to elucidate the succession of organic matter
sources/composition from the fresh water to the estuarine sections of the Rajang. Our aim was to
address the following questions: 1) Given that peatland contributes additional DOC to fluvial DOC
(Müller et al., 2016), does the composition of dissolved OM change from river to estuary? 2) Do
changes in organic nitrogen mirror changes in organic carbon? 3) And hence what is the role of
peatland/mangroves on OM composition and lability in the Rajang? Globally, rivers in low latitudes
receive much less attention relative to temperate and polar rivers (36 vs. 958 studies)(Cloern et al.,
2014), while they could equally important in carbon cycle (Cloern et al., 2014). Our work, together
with other tropical studies, would enrich the understandings for organic carbon and nitrogen cycles
in tropical rivers/estuaries.

**2. Materials and methods**
All abbreviations, together with the amino acids measured in this study, are listed in table 1.
**2.1 Brief background**
The Rajang river and estuary is located in Sarawak, Malaysia. The climate is wet year-round,



but the main precipitation typically occurs in winter (November to February). Climate is influenced
by El Niño-Southern Oscillation (ENSO) and Madden-Julian Oscillation. In August 2016, the
discharge was estimated as 2440 m$^3$/s, in comparison with an annual mean discharge of 4000 m$^3$/s
for 2016 and 2017 (Müller-Dum et al., 2019).

Based on salinity, Sibu is regarded as the boundary of the fresh and estuarine water of the

Rajang (Fig. 1b). In this work all samples with a salinity of 0 were regarded as fresh water, while
samples with salinity >0 were regarded as estuarine. In the estuary, there are several branches,
namely Igan, Lassa, Paloh, and Rajang itself (Fig. 1b). Since water in all these branches are from
Rajang river (i.e., upstream of Sibu), in this work all these branches are regarded as the Rajang
estuary. Peatland and mangroves are commonly distributed in the estuary (shown in Fig. 1b) while
tropical rainforest is widely distributed upstream of Sibu (not shown in Fig. 1b). The peatland is
under strong pressure of draining and change of use for oil palm, while in the basin logging and
secondary growth is very common (Hooijer et al., 2015). Compared with other peatland-draining
tropical blackwater rivers, the Rajang is more like a turbid tropical rainforest river (Müller-Dum et
al., 2019), but with notable peatland/mangrove in its estuary (Fig. 1b).
**2.2 Field sampling**

The field work was carried out in August 2016. The sampling stations covered from Kapit (the

upper most station in this study) to S1 on the coast. At each station, a pre-cleaned and sample-rinsed
bucket was used to collect surface water from the center of the channel in a boat. After sample
collection, pretreatment was done immediately on board in the boat. For DOC and its stable carbon
isotope ratios ($\delta^{13}C$), water samples were collected by syringe filtering (pre-combusted Whatman
GF/F; 0.7 μm) approximately 30 ml of sample water into a pre-combusted 40 ml borosilicate vial.



Samples were preserved with five drops of concentrated phosphoric acid and sealed with a lid
containing a Teflon-coated septa. For total dissolved amino acids (TDAA), water samples were
filtered through a 0.4 μm nylon filter. For particulate OM samples (TSM, POC, POC-$\delta^{13}$C, PN and
PN-$\delta^{15}$N, and total particulate amino acids (TPAA)), suspended particles were concentrated onto
glass fiber membrane (pre-combusted Whatman GF/F; 0.7 μm). The GF/F filters were folded and
packed in pre-combusted aluminum. All samples were immediately stored frozen (-20°C) until
analysis. A portable meter (Aquaread, AP-2000) was used to obtain conductivity/salinity,
temperature, dissolved oxygen and pH.
**2.3 Laboratory analyses**

Concentrations and $\delta^{13}$C of DOC were measured via continuous-flow wet oxidation isotope-

ratio mass spectrometry using an Aurora 1030W total organic carbon analyzer coupled to a Thermo
Delta V IRMS (Oakes et al. 2010). Glucose of known isotopic composition dissolved in He-purged
Milli-Q was used as a standard to correct for drift and to verify sample concentrations and $\delta^{13}$C
values. Reproducibility for concentrations and $\delta^{13}$C was ± 0.2 mg l$^{-1}$ and ± 0.4 ‰. DOC
concentrations and $\delta^{13}$C were measured at the Centre for Coastal Biogeochemistry at Southern Cross
University (Lismore, Australia). For the determination of POC, samples (GF/F glass fiber filter)
were freeze-dried and analyzed with a CHNOS analyzer (Model: Vario EL III) after removing the
inorganic carbon by reaction with HCl vapor. For PN, a similar procedure like POC was followed,
but no acid was used in pre-treatment. The detection limit for POC was 7.5×10$^{-6}$ g, with precision
better than 6%, based on repeated determinations (Zhu et al., 2006). The POC-$\delta^{13}$C and PN-$\delta^{15}$N
were determined using a DELTAplus/XL isotopic ratio mass spectrometer (Finnigan MAT Com.
USA) interfaced with a Carlo Erba 2500 elemental analyzer. The standard for $\delta^{13}$C was PDB and



the precision of the analysis was ± 0.2‰. For $\delta^{15}N$, the standard was air and precision was ± 0.3‰.
Total hydrolyzable AAs were extracted and analyzed following the method of Fitznar et al., (1999)
with slight modifications (Zhu et al., 2014). Briefly, samples were first hydrolyzed with HCl at
110°C. After pre-column derivatization with o-Phthaldialdehyde (OPA) and N-Isobutyryl-L/D-
cysteine (IBLC/IBDC), AAs and their enantiomers were analyzed using an HPLC (Agilent 1200)
comprising of an online vacuum degasser, a quaternary pump, an auto-sampler, a thermostatted
column and a fluorescence detector (excitation 330 nm, emission 445 nm). The analytical column
was a Phenomenex Hyperclone column (BDS C18, 250×4mm, 5μm) with a corresponding pre-
column. To eliminate the influence of racemization of L-type AAs in the hydrolysis process, the
concentration of D/L AAs measured in actual samples was corrected according to the formula
obtained by Kaiser and Benner (2005). The detection limit for glycine (Gly) and individual AAs
enantiomers were in the lower picomolar level. Asx and Glx were used for aspartic acid + asparagine
and glutamic acid + glutamine, respectively (Table 1), as the corresponding acids are formed via
deamination during hydrolysis.
A few samples (e.g., TDAA in S1 station) were not measured due to instrument hardware problem.
And hence the measured particulate and dissolved sample stations did not exactly match.

**3. Results**
In August 2016, the TSM concentration in the Rajang ranged from 22 mg/L (mean for the fresh
water section) to 161 mg/L (mean for the estuarine section) (Table 2). Throughout the system DOC
concentrations exceeded POC concentrations. DOC and POC in the fresh water section averaged
337 μM and 86 μM, and in the estuarine section 345 μM and 64 μM, respectively (Table 2). While



DOC concentration was slightly higher in the estuary than in the fresh water (Table 2), a maximum
of both DOC and POC can be found at around salinity 15 to 20 in the estuary (Fig. 2).

DOC $\delta^{13}$C ranged from -28.7‰ to -20.1‰ (Table 2). A U-shaped trend from fresh water

section to estuary section can be identified for DOC $\delta^{13}$C, with one outlier from the Rajang main
stream at a salinity of 5 (S2 station; Fig. 3a). The minimum value of DOC $\delta^{13}$C (bottom of the U)
was detected at a salinity of ~10 (Fig. 3a). For particulate OM, $\delta^{13}$C ranged between –29.4‰ to –
31.1‰ in the fresh water section. In the estuary section, there was a clear increasing trend with
increasing salinity, from –30‰ (S=1.1) to values close to –24‰ (S>30) (Fig. 3b).

In the fresh water section, the mean TDAA and TPAA concentrations were 0.3 µM and 2.5

µM, respectively (Table 3). For TDAA, the AA carbon yield (the carbon from AA divided by bulk
DOC or POC, in %) in both fresh water and estuary sections were very similar, namely 0.40% and
0.38% (mean), respectively (Table 3), whereas AA nitrogen yield was higher in the estuary (11%)
than in the fresh water section (4.8%) (Table 3). For TPAA, there was little difference between the
fresh water and estuary sections in AA carbon yield (13.5% and 16.8%, respectively) and nitrogen
yield (66% and 62%, respectively) (Table 3).

With respect to individual AA compounds, in both dissolved and particulate phase, Gly, Glx,

Ala and Asx were the most abundant four AAs. These four AAs together accounted for 66% of
TDAA and 47% of TPAA in the fresh water section, 59% of TDAA and 48% of TPAA in the estuary.
The non-protein AA GABA was detected in trace amounts, but was accumulated in the dissolved
phase relative to the particulate phase, as indicated by the higher GABA% in the dissolved phase
(Table 3). GABA% decreased from 2% (fresh water section mean) to 1.3% (estuarine section mean)
in the dissolved phase, and decreased from 0.7% (fresh water section mean) to 0.4% (estuarine





section mean) in the particulate phase (Table 3). In the estuary, GABA% in the dissolved phase
remained stable (~1.5%) in brackish water (salinity 5 to 20) and quickly dropped to <1% where
salinity was over 30 (Fig. 4a). Most of the GABA% dots were above the theoretical dilution line
(Fig. 4a). In the particulate phase, there was an overall decrease in GABA% with increasing salinity
within the estuary (Fig. 4b).

As for the AA enantiomers, the D/L ratio of AA in the dissolved phase averaged 12% for both

fresh water and estuarine section. The most abundant D-form AAs in the dissolved phase were Glx
and Asx. For the particulate phase, the D/L ratio of AA was much lower, decreasing from a mean
of 4.4% in the fresh water section to a mean of 3.3% in the estuary (Table 3). And patterns in the
variation of D/L Glx (Fig. 5) along with conductivity/salinity gradient in the Rajang were similar to
those for GABA% (Fig. 4) for both dissolved and particulate phase. For example, for dissolved
phase, a similar platform can be identified at salinity range of 5 to 20 (Fig. 5a), whereas for
particulate phase the decreasing pattern along with salinity is very clear in the estuary (Fig. 5b).
Also, for dissolved phase in the estuary, all the data were above the theoretical dilution line for D/L
Glx.

**4. Discussion**
**4.1 Distribution patterns of OM composition**
*Dissolved OM*

Terrestrial OM usually has a more negative $\delta^{13}C$ value (–32‰ to –26‰ for C3 plants), whereas

marine OM has more positive value values ($\delta^{13}C$, ~ –20‰) values (Lamb et al., 2006; Mayorga et
al., 2005). Overall, the very negative $\delta^{13}C$ values for DOC (<–26‰) in the river part of the Rajang





indicates that the OM had a very clear C3 plant source (e.g., mangroves and oil palms (Jennerjahn
et al., 2004; Lamade et al., 2009; Wu et al., 2019)), whereas DOC $\delta^{13}$C values > –24‰ in the estuary
(salinity >30) suggest a mixture of terrestrial and marine OM (Fig. 3a). The most depleted $\delta^{13}$C
values for DOC occurred at a salinity of 10 (Fig. 3a). Above this salinity, the influence of marine
OM became more overwhelming, and the bulk DOC $\delta^{13}$C signal was more enriched (Fig. 3a).
Among samples in the fresh water section, the sample of most enriched DOC-$\delta^{13}$C value (S10
and S15; DOC-$\delta^{13}$C: –25‰; Fig. 3a) although initially appearing to be outliers, were characterized
by very elevated D/L amino acids ratios (Fig. 6a). This was particularly the case for the sample from
S10 (the upper most station in this study; Fig. 1b), which showed a maximum D/L Glx ratio of 0.57
(Fig. 6a). In addition, these samples from S10 and S15 also showed a higher D/L ratio for Asp (S10:
0.49, S15: 0.38; figure not shown) when compared to all fresh water or estuary samples (mean: 0.34;
Table 3). On land, D form amino acids can be derived from abiotic racemization process (which
requires a very long time scale) by which L form amino acids slowly changed into their
corresponding D form (Schroeder and Bada, 1976). More significantly in contemporary
environments, D form amino acids are widely synthesized by bacteria during their cell membrane
construction (Schleifer and Kandler, 1972). D/L Glutamic acid and D/L Aspartic acid ratios of pure
peptidoglycan (*Staphylococcus aureus*, Gram-positive) are 0.49 and 0.30, respectively (Amon et al.,
2001). Though $\delta^{13}$C values for bacteria in the Rajang remains unclear, bacteria have been reported
to have $\delta^{13}$C values from –12‰ to –27‰ (Lamb et al., 2006). Contribution of OM derived from
bacteria may therefore explain the relatively enriched $\delta^{13}$C values observed at inland S10 station
and S15. A possible OM source at these stations is soil humic substances, which are expected to be
under strong impact of bacteria, and have a high contribution of D-form amino acids (Dittmar et al.,





2001a). A more depleted pattern of DOC $\delta^{13}$C from mountain to lowland is suggested to be due to
dilution and mixing with younger OM in the lowland (Mayorga et al., 2005). This is consistent with
our findings that, depleted pattern of riverine DOC $\delta^{13}$C within the fresh water section was
corresponding to a lowering D/L ratio pattern, which indicates the dilution with less degraded OM
(Fig. 6a). Whether the dissolved samples with elevated D/L ratio and relatively positive $\delta^{13}$C in the
fresh water section (S10 and S15; Fig. 6a) reflect the presence of soil humic substances, or instead
reflect the direct presence of bacteria, requires further study.

In the estuarine section, it was very clear that terrestrial bio-degraded OM (indicated by

elevated D/L ratios and more negative $\delta^{13}$C) is diluted with more labile OM (lower in D/L ratio but
more positive $\delta^{13}$C)(Fig. 6a). However, this apparent dilution trend became very vague (or showed
no trend) when D/L ratio was plotted against salinity (Fig. 5a). This was also confirmed by the
GABA% distribution pattern which showed a platform-like pattern at a salinity between 5 and 20
(Fig. 4a). Though TDAA at S1 is missing, the composition of TDAA at S2 (salinity = 31.2) was
very typical of marine OM (i.e., very low D/L ratio and relatively enriched DOC-$\delta^{13}$C; see Fig. 6a).
Hence in the estuary there is a conservative distribution pattern for dissolved OM when plotted
against $\delta^{13}$C (Fig. 6a) but such pattern disappeared when plotted against salinity (Fig. 4a&5a). The
location above the conservative dilution line of all OM data in the brackish estuary (salinity between
10 and 25; Fig. 4a&5a), indicates that the OM in the estuarine section was more degraded than
theoretically expected. The combination of degraded OM with the observed DOC concentration
increase in the estuary (345 μM in the estuary vs. 337 μM in the fresh water section; or Fig. 2b),
suggests the addition of degraded DOC to the Rajang. Non-conservative dissolved OM behavior in
the estuary has previously been reported based on an optical approach (Martin et al., 2018), and





minimal OM alteration during estuarine transport was suggested (Martin et al., 2018). Hence, it is
reasonable that changes in dissolved OM composition (Fig. 4a&5a) may largely take place in
land/estuary (e.g., in pore waters of soil) and impact the Rajang riverine dissolved OM via leaching
from soils.
*Particulate OM*

As for dissolved OM, depleted POC-$\delta^{13}$C in the river part of the Rajang indicated the strong

influence of terrestrial OM (e.g., C3 plantDittmar et al., 2001b) whereas in the estuary, particulate
OM was diluted with marine particulate OM, as indicated by the seawards enrichment of $\delta^{13}$C (Fig.
3b). In the sediment, a clear woody angiosperm C3 plants as the OM source is found based on a
lignin approach (Wu et al., 2019), and similar increases in carbon and nitrogen isotopes in suspended
particles in brackish water have also been observed in other estuaries (Cifuentes et al., 1996;
Raymond and Bauer, 2001). Unlike dissolved OM, there were no samples with unusually enriched
$\delta^{13}$C values in the fresh water section (Fig. 6b&c). D/L Glx ratio in the fresh water section is higher
when compared with that in the estuary section (Table 3), and overall, when compared with
dissolved OM, particulate OM basically became more labile when transporting seawards, as
indicated by its composition shift along with salinity (Fig. 4b&5b) or isotope (Fig. 6b&c).

Although particulate OM had a lower D/L ratio than dissolved OM (Fig. 6), it should be noted

that this does not mean dissolved OM is more aged or degraded than particulate OM. Rather, as
observed in other estuaries (Dittmar et al., 2001a), bacteria and their detritus simply tend to
accumulate in the dissolved phase, relative to the particulate phase.
**4.2 Different fate of bulk organic carbon and nitrogen**

Leaching of DOC and DON from peatlands is driven by difference mechanisms; whereas DOC





release is related to the status of peatland (pristine vs. degraded), DON release is determined by the
DIN content of peatland soil (Kalbitz and Geyer, 2002). In the Rajang, bulk DOC and DON
concentrations were not coupled, as suggested by the DOC/DON ratio variation pattern (Fig. 7).
The average DOC concentration in the estuary part was slightly higher (345 µM) than in the river
part (337 µM; Table 3), which indicates the addition of DOC in the estuary. In comparison, the
removal of DON in the estuary is suggested (Jiang et al., 2019).

In the Rajang, non-conservative dilution behavior from optical properties was observed for

estuarine DOC (Martin et al., 2018), which is consistent with other peatland-draining rivers in
Sarawak (Müller et al., 2016). The contribution of marine sources to dissolved OM is reflected in
the increasing DOC-$\delta^{13}$C in the estuary part (Fig. 3a). Peatland, however, is known for its high
contribution to fluvial DOC and has been suggested to contribute to the DOC in the Rajang (Martin
et al., 2018). In peatland-draining rivers west of the Rajang, the DOC concentration endmember can
be as high as 3690 µM (Müller et al., 2015). Under such high DOC background, a simple three-
point mixing model (i.e., a model that based on 1 observed fresh water DOC endmember, 2 peatland
DOC endmember and 3 calculated fresh water DOC endmember) suggests that peatland-DOC
addition accounts for 3% of the fluvial DOC in the Saribas river and 15% in the Lupar river (Müller
et al., 2016). Assuming that peatland in the Rajang estuary has a comparable endmember DOC
concentration to other peatland in Sarawak (i.e., 3690 µM; Müller et al., 2015), and given our
observed Rajang fresh water DOC endmember value of 337 µM (DOC concentration at S5 station)
and a marine DOC endmember of 238 µM (S1 station), a similar model approach suggests peatland
DOC addition contributed 4% of the Rajang fluvial DOC, which is comparable to Saribas river and
much lower than Lupar river (Müller et al., 2016). In the meantime, as mentioned in the previous





section, there is a non-conservative dilution pattern, with dissolved OM in the estuary part more
degraded than expected based on simple dilution with a marine endmember (Fig. 4a&5a). Hence it
is reasonable that peatland not only contributed to the fluvial DOC in concentration (Martin et al.,
2018), but also modified the dissolved OM composition (more bio-degraded) in the estuary. In
another tropical river study, mangrove in the estuary exerted a stronger influence on fluvial
dissolved OM than hinterland vegetation (Dittmar et al., 2001b). This is consistent with the Rajang,
for which estuarine processes apparently impact the dissolved OM in terms of both DOC
concentration (by increasing the bulk amount) and composition (by adding bio-degraded DOC). The
estuarine dissolved OM showed higher bio-degraded feature (e.g., elevated GABA% and D/L ratio;
Fig. 4a&5a), but this subpart may be photolabile (Martin et al., 2018). When TSM decreases and
light condition in the water column becomes good (e.g., entering the sea), photodegradation is
expected (Martin et al., 2018). Other oceanic degradation mechanisms include the priming effect
(Bianchi, 2011). The fate of the terrestrial OM in the sea requires further study. As we lack the DON
concentration endmember in peatland, peatland impact on DON in the estuary is not estimated.

In contrast to DOC, which was apparently added to the estuary, DON was removed,

contributing to a remarkable increase of dissolved inorganic nitrogen in the estuary (Jiang et al.,
2019). In the fresh water section, the nitrate concentration was not related to the ratio of D/L
dissolved AAs, nor related to dissolved GABA% (Fig. 8), and in the estuarine section, nitrate was
not related to D/L AAs but it indeed was related to GABA% in the estuarine section (Fig. 8b). This
indicates that fluvial nitrate in the fresh water section was not derived from remineralization of
fluvial organic matter in the river channel, but more likely from other sources (e.g., leaching of soil).
In the estuarine section, there may be some DON transformation occurred (Jiang et al., 2019), while



the leaching from soil process still cannot be eliminated (Fig. 8). For particulate phase, no relation
can be found between nitrate and particulate OM composition (figure not shown).
The atomic DOC/DON in Rajang averaged 50 in the river part, and increased to 140 (mean
value) in the estuary part (Fig. 7). Although the DOC/DON ratio was much higher when compared
to other tropical peatland river waters (around 10; Sjögersten et al., 2011), the ratio is comparable
with other peatland-draining rivers in Sarawak like the Lupar, Saruba and Maludan rivers (Müller
et al., 2015; Müller et al., 2016), which all enter the South China sea. The ratio is also within the
reported C/N ratio of peatland and leaves (Müller et al., 2016). For the Amazon river, the DOC
versus total nitrogen ratio ranges from 27 to 52 (Hedges et al., 1994). Given their reported total
nitrogen includes inorganic nitrogen, the DOC/DON ratio for the Amazon river would be even
higher. Under the background of such high C/N ratios (e.g., 50), transformation of DON to DIN in
the estuary further enhanced the high DOC/DON ratio (to 140), and hence a deficiency in terrestrial
organic nitrogen output is expected. We noted that dissolved inorganic nitrogen for Rajang is on the
order of 10 μM, comparable to DON (Jiang et al., 2019). Terrestrial nitrogen output is an important
source for coastal primary production (Jiang et al., 2019), but peatland-impacted rivers may have
relatively lower nitrogen input to the South China Sea when compared with their very high river
basin DOC yields (Baum et al., 2007). On one hand, logging and secondary growth has been found
to play a negative role in the nitrogen output efficiency of forest soils (Davidson et al., 2007). On
the other hand, disturbed tropical peatlands could release more DOC in comparison to an
undisturbed site (Moore et al., 2013) while the DOC/DON ratio may also decrease along with
disturbance of peatland (Kalbitz and Geyer, 2002). Given that secondary growth in river basin and
anthropogenic disturbance of peatland (e.g., drainage and conversion for oil palm) are both common





(Hooijer et al., 2015), changes of DOC/DON ratios in the Rajang are complex and further
assessment is needed in the future.

**5. Summary and Conclusion**

In August 2016 in the Rajang, we observed that dissolved OM composition (as D/L Glx ratio)

was conservatively diluted along with increasing DOC $\delta^{13}$C, indicating that the sources of dissolved
OM have a very conservative impact on the OM composition. When D/L Glx ratio was plotted
against salinity (as is usually done for an estuarine OM behavior check in many studies), such linear
conservative dilution pattern disappeared (Fig. 4a&5a). This implies that the addition of DOC in the
estuary (peatland/mangrove) had an impact on dissolved OM composition, adding more bio-
degraded OM, and resulting in data above the theoretical dilution line (Fig. 4a&5a). For particulate
OM, though the data was variable, the overall decreasing GABA% or ratio along with salinity was
much clearer relative to that of dissolved OM (Fig. 4b&5b). Particulate D/L Glx ratio in the estuary
was usually lower when compared with that in the fresh water section (Fig. 6b&c), whereas for
dissolved OM, the majority of the samples in the estuary had a D/L Glx ratio similar to that in the
fresh water (Fig. 6a). The difference in OM composition between fresh water and estuarine section
suggests that dissolved OM became more degraded while particulate OM became less degraded in
the estuary.

The Rajang is characterized by DOC/DON ratios of 50 in the fresh water section, and the

further loss of DON in the estuary increased the ratio to 140. Peatland draining and
logging/secondary growth are reported to have conflicting impacts on carbon and nitrogen cycling
(Davidson et al., 2007; Moore et al., 2013), which may increase fluvial DOC and limit basin nitrogen



output, resulting in even larger DOC/DON. Mismatch in carbon and nitrogen loss from tropical
rivers due to anthropogenic activities plays a role in material cycle in both land and marine systems,
enhancing the tropical river as a direct carbon source to atmosphere while for nitrogen change and
its further feedback on carbon cycle needs further monitoring and assessment.
At last, this work is based on a dry season investigation (August). Though the seasonality for
Rajang OM may be moderate (Martin et al., 2018), for biomarkers like amino acids enantiomers
further investigation in the wet season is needed.

**Acknowledgements**
We thank captain and crew of the boat, as well as other colleagues on board during the field
work. We thank colleagues and students in both Swinburne University of Technology (Sarawak
Campus) and in State Key Lab of Estuarine and Coastal Research/East China Normal University.
We thank Aazani Mujahid in University Malaysia Sarawak for her help and hospitality during our
stay in Malaysia. This work is funded by the National Key Research and Development Program of
China (2018YFD0900702), MOHE FRGS 15 Grant (FRGS/1/2015/WAB08/SWIN/02/1) in
Malaysia, ARC Linkage Grant LP150100519 in Australia, a SKLEC Open Research Fund (SKLEC-
KF201610) and '111' project in SKLEC/ECNU from the Ministry of Education of China and State
Administration of Foreign Experts Affaires of China.

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





Table 1. Measured amino acids (the L- and D- enantiomers are not listed) and all abbreviations in
this study. Note that glycine has no enantiomer.

| name | abbreviations |
|---|---|
| organic matter | OM |
| dissolved organic carbon | DOC |
| dissolved organic nitrogen | DON |
| total suspended matter | TSM |
| amino acid | AA |
| total hydrolysable dissolved amino acids | TDAA |
| total hydrolysable particulate amino acids | TPAA |
| Alanine | Ala |
| Arginine | Arg |
| Asparagine | Asx |
| Aspartic acid | |
| Glutamine | Glx |
| Glutamic acid | |
| Glycine | Gly |
| Isoleucine | Ile |
| Leucine | Leu |
| Lysine | Lys |
| Methionine | Met |
| Phenylalanine | Phe |
| Serine | Ser |
| Threonine | Thr |
| Tryptophan | Trp |
| Tyrosine | Tyr |
| Valine | Val |
| γ - aminobutyric acid | GABA |






Table 2. TSM, DOC, POC and stable carbon isotopes in the freshwater and estuary of the Rajang
(mean (min-max)).

|  | unit | Fresh water | Estuary |
|---|---|---|---|
| TSM | mg/L | 61 (22 - 126) | 73 (25 - 161) |
| DOC | μM | 337 (217 - 658) | 345 (214 - 587) |
| DOC $\delta^{13}$C | ‰ | −26.7 (−27.7 - −25.0) | −26.1 (−28.7 - −20.1) |
| POC | μM | 86 (46 - 125) | 64 (22 - 153) |
|  | % | 1.9 (1.2 - 2.5) | 1.0 (0.6 - 1.9) |
| POC $\delta^{13}$C | ‰ | −30.1 (−31.1 - −29.4) | −26.7 (−30.1 - −23.8) |

Table 3 The Rajang AAs result (mean (min-max)) in August 2016 (*total D/TDAA means total D
form AA versus TDAA, the same for total D/TPAA)

|  |  | unit | Fresh water | Estuary |
|---|---|---|---|---|
| dissolved | TDAA | nM | 317 (131 - 486) | 523 (212 - 2320) |
|  | TDAA carbon yield | % | 0.40 (0.08 - 0.65) | 0.38 (0.29 - 0.53) |
|  | TDAA nitrogen yield | % | 4.8 (1.3 - 15) | 11 (5.4 - 18) |
|  | GABA | % | 2.0 (1.3 - 4.1) | 1.3 (0.15 - 1.9) |
|  | total D/total TDAA* | % | 12 (8 - 15) | 12 (3 - 14) |
|  | D/L Glx |  | 0.35 (0.16 - 0.57) | 0.32 (0.07 - 0.42) |
|  | D/L Asx |  | 0.34 (0.23 - 0.48) | 0.34 (0.08 - 0.42) |
| particulate | TPAA | μM | 2.5 (1.4 - 3.6) | 2.0 (1.1 - 3.7) |
|  | TPAA carbon yield | % | 14 (9.5 - 19) | 17 (11 - 24) |
|  | TPAA nitrogen yield | % | 66 (36 - 82) | 62 (30 - 100) |
|  | GABA% | % | 0.7 (0.6 - 0.9) | 0.4 (0.2 - 0.8) |
|  | total D/total TPAA* | % | 4.4 (3.6 - 5.2) | 3.3 (2.4 - 5.0) |
|  | D/L Glx |  | 0.09 (0.08 - 0.10) | 0.06 (0.04 - 0.08) |
|  | D/L Asx |  | 0.04 (0.03 - 0.05) | 0.05 (0.03 - 0.11) |



## Figure caption

Figure 1. Study area and sampling stations. a) Location of Sarawak, Malaysia; and b) the Rajang with its estuary/river mouth background shown. Samples upstream of Sibu showed 0 salinity while downstream of Sibu showed salinity >0. Hence from Sibu to Kapit is regarded as the fresh water section, and downstream of Sibu is regarded as the estuarine section.

Figure 2. Distribution pattern of (a) TSM, (b) DOC and (c) POC along with conductivity/salinity in the Rajang. The location of salinity = 0 is at Sibu (Fig. 1b). The legend indicates the branches that the samples were from and marine corresponds to S1 station.

Figure 3. Distribution pattern of (a) DOC $\delta^{13}$C and (b) POC $\delta^{13}$C along with conductivity/salinity in the Rajang. The legend indicates the branches that the samples were from and marine corresponds to S1 station.

Figure 4. GABA% distribution pattern from fresh water to estuary in the Rajang: a) dissolved and b) particulate. The legend indicates the branches that the samples were from and marine corresponds to S1 station.

Figure 5. D/L ratio of Glx from fresh water to estuary in the Rajang: a) dissolved and b) particulate. The legend indicates the branches that the samples were from and marine corresponds to S1 station.

Figure 6. D/L ratio of AAs (as Glx) plotted against a) DOC $\delta^{13}$C b) POC $\delta^{13}$C, and c) PN $\delta^{15}$N.

Figure 7. DOC/DON ratio distribution pattern along with salinity in the Rajang. For fresh water and estuary, the mean DOC/DON value was 50 and 140, respectively. DON is from Jiang et al., (2019)

Figure 8. Dissolved OM composition (a: D/L Glx, b: GABA%) and its relation with nitrate. Nitrate is derived from Jiang et al., (2019).

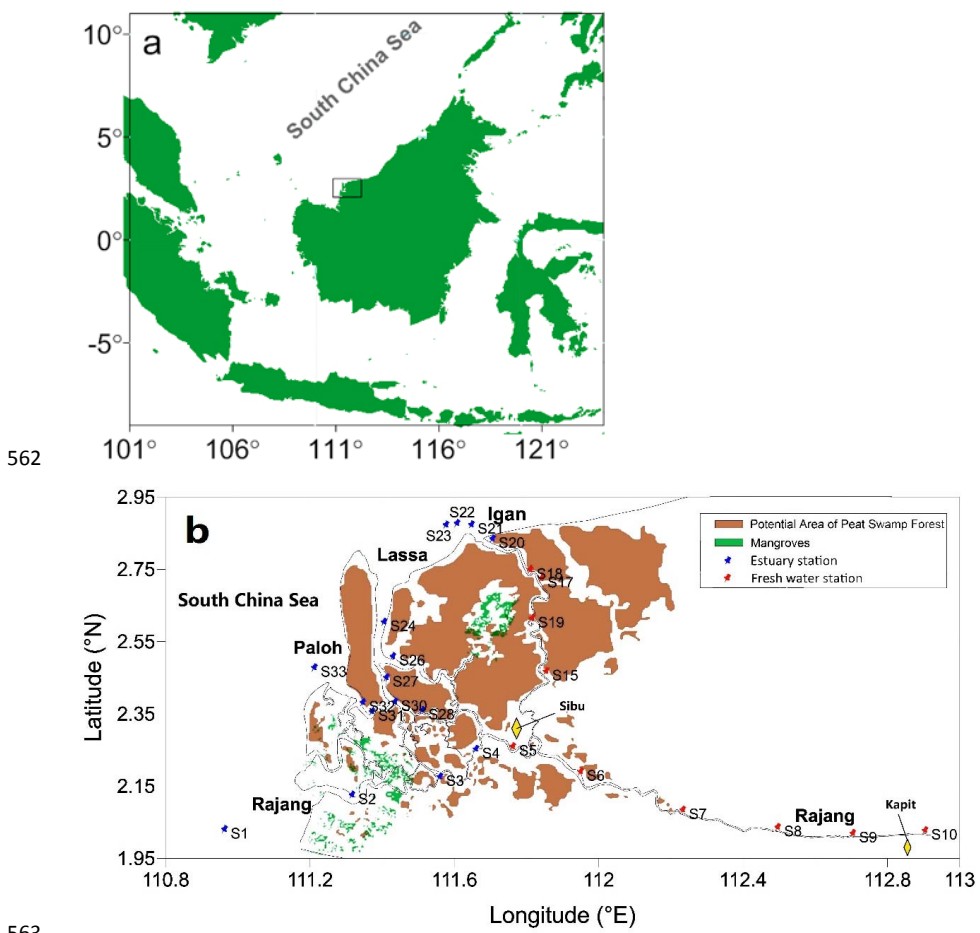

Figure 1. Study area and sampling stations. a) Location of Sarawak, Malaysia; and b) the Rajang
with its estuary/river mouth background shown. Samples upstream of Sibu showed 0 salinity while
downstream of Sibu showed salinity >0. Hence from Sibu to Kapit is regarded as the fresh water
section, and downstream of Sibu is regarded as the estuarine section.

.



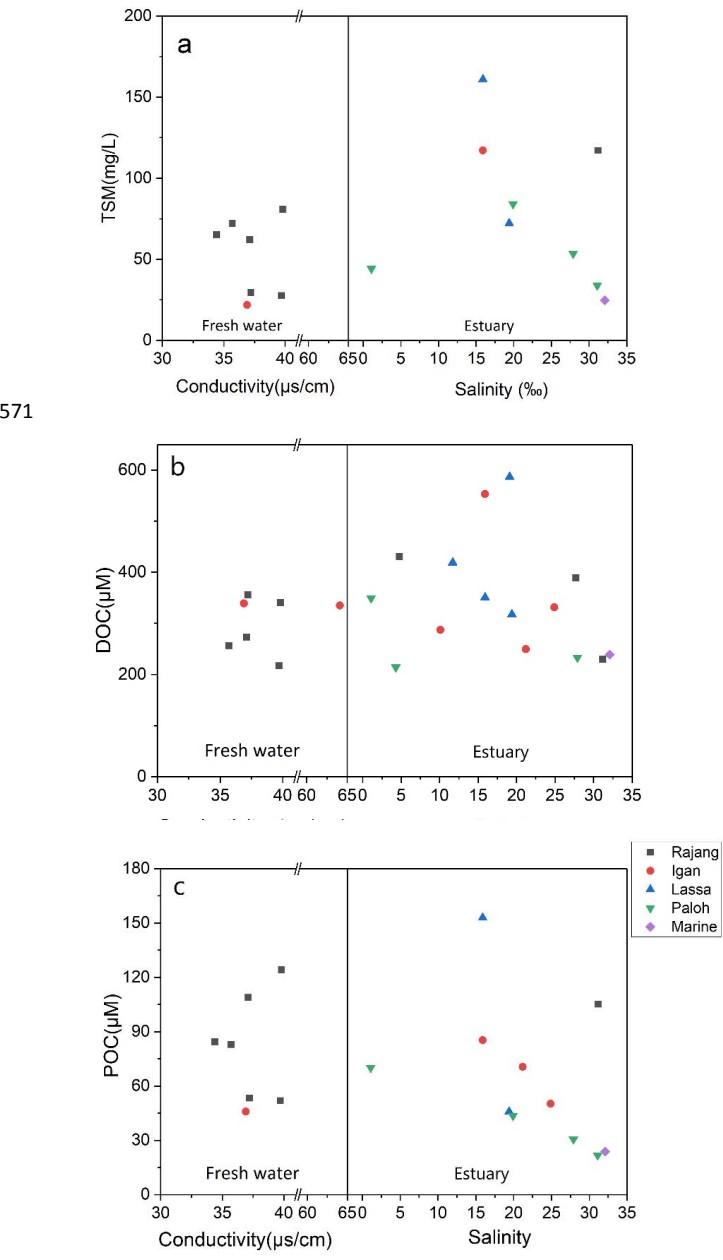



Figure 2. Distribution pattern of (a) TSM, (b) DOC and (c) POC along with conductivity/salinity
in the Rajang. The location of salinity = 0 is at Sibu (Fig. 1b). The legend indicates the branches
that the samples were from and marine corresponds to S1 station.



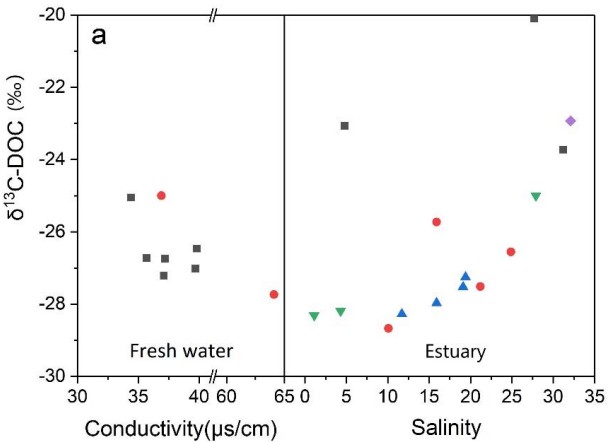

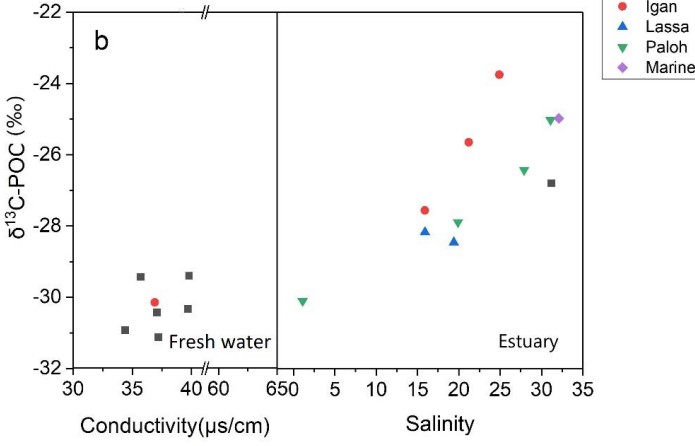


Figure 3. Distribution pattern of (a) DOC $\delta^{13}$C and (b) POC $\delta^{13}$C along with conductivity/salinity
in the Rajang. The legend indicates the branches that the samples were from and marine
corresponds to S1 station.





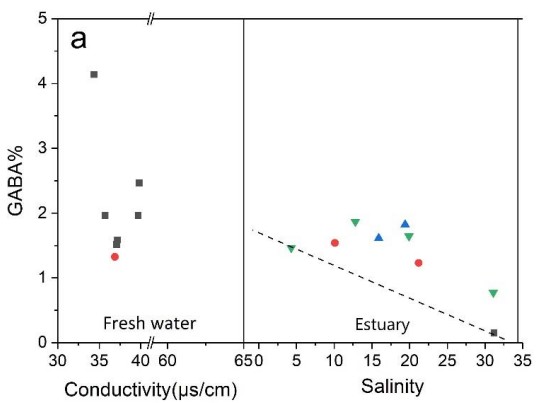

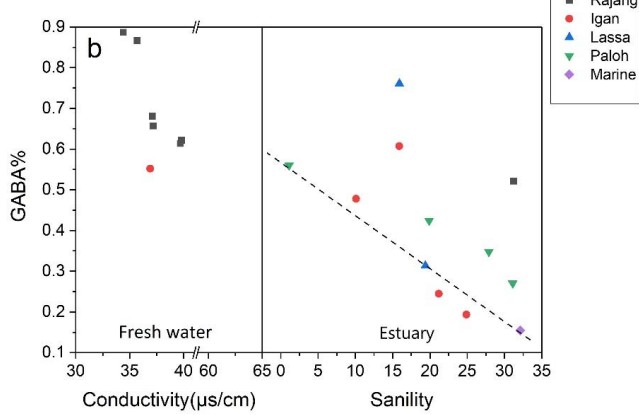


Figure 4. GABA% distribution pattern from fresh water to estuary in the Rajang: a) dissolved and
b) particulate. The legend indicates the branches that the samples were from and marine
corresponds to S1 station.


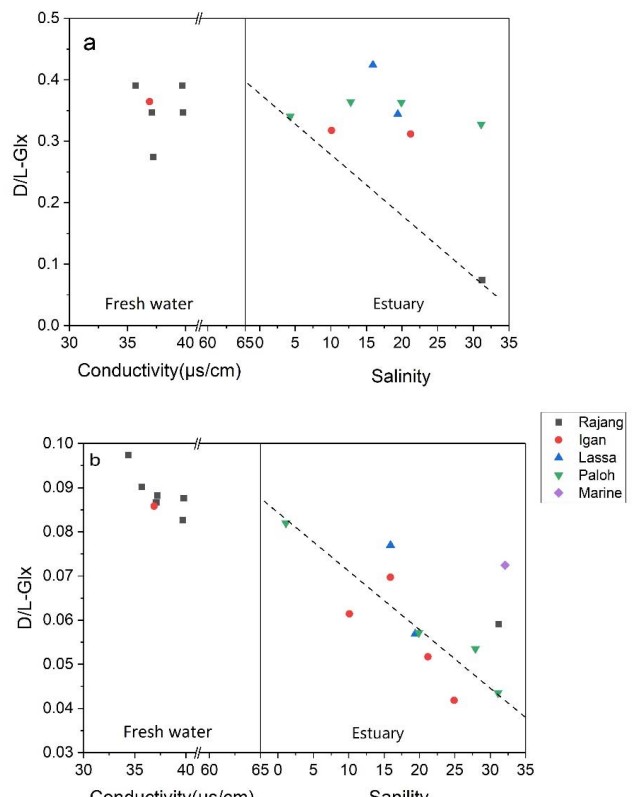



Figure 5. D/L ratio of Glx from fresh water to estuary in the Rajang: a) dissolved and b) particulate.
The legend indicates the branches that the samples were from and marine corresponds to S1
station.







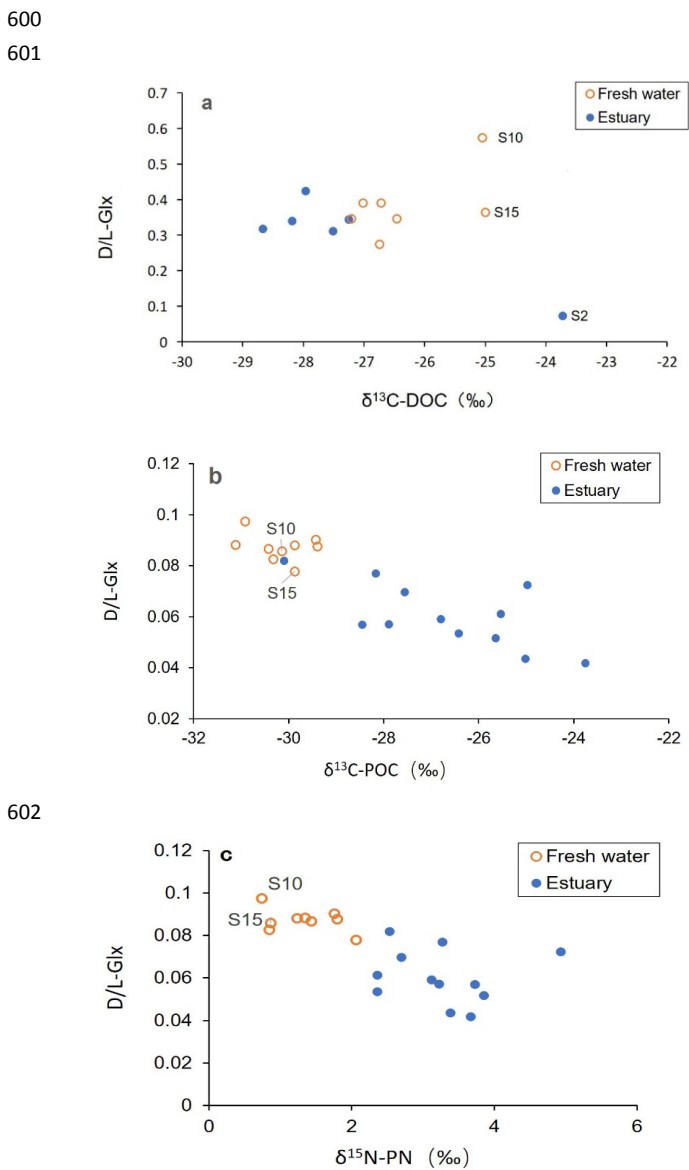



Figure 6. D/L ratio of AAs (as Glx) plotted against a) DOC $\delta^{13}$C b) POC $\delta^{13}$C, and c) PN $\delta^{15}$N.







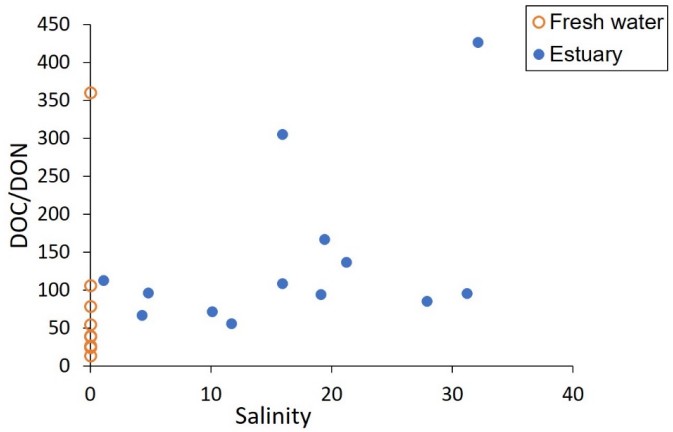


Figure 7. DOC/DON ratio distribution pattern along with salinity in the Rajang. For fresh water
and estuary, the mean DOC/DON value was 50 and 140, respectively. DON is from Jiang et al.,

(2019)









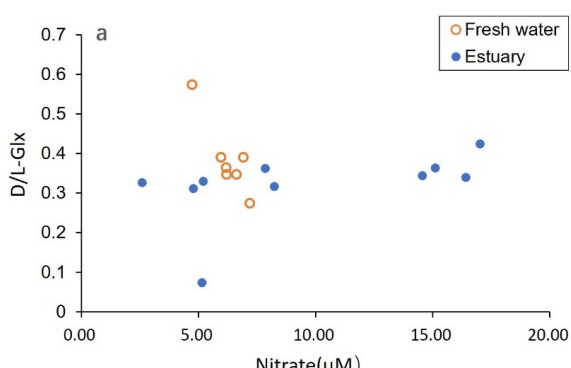


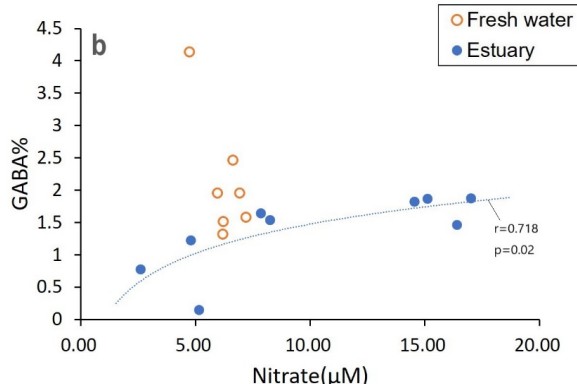


Figure 8. Dissolved OM composition (a: D/L Glx, b: GABA%) and its relation with nitrate. Nitrate

is derived from Jiang et al., (2019).

