# Peer review of "The non-conservative distribution pattern of organic matter in"

_Biogeosciences, 2019_

## Author Comment (AC1) · 25 Aug 2019

We think the paragraph from line 294 to line 297 was a little bit confusing, especially the words "bacteria and their detritus simply tend to 297 accumulate in the dissolved phase, relative to the particulate phase" literally is misleading. —We should say that bacteria tends to attach to particles, relative to dissolved form.

When judged by biomarker signals like D/L ratios of amino acid enantiomers (fig. 6), DOM and POM had ratios of ∼0.5 and 0.1, respectively (fig. 6). On one hand, by comparing the ratios itself, it seems that DOM is highly degraded/more refractory relative to POM (as higher D/L ratio usually indicates more refractory and 0.5>0.1). And bacteria tends to attach to the particles while its metabolism products (which may be more refractory) tends to be released as dissolved form, which also indicates that DOM may be more refractory. On the other hand, though the age of POM and DOM is not available in this work, but in river/estuary the dissolved OM usually is younger relative to particulate OM (i.e., DOM usually has an older 14C age than POM), which complicates the comparison of OM natures. Hence, we should be very careful in making any conclusions in comparing the POM and DOM lability/refractory nature in the river/estuary. By only D/L ratio along it is hard to reach a solid conclusion which form (dissolved vs. particulate) is more refractory.

---

## Short Comment (SC1) · 8 Oct 2019

A very impressive paper about the OM composition modification at a tropical river-estuary system!

Although the $\delta$13C of DOC indicated a significant conservative behavior of DOC at this estuary, detailed OM compositions surprisingly uncovered the nonconservative distributions of DOM.

Given this estuary is surrounded by peatland, the addition of DOC from peatland is reasonable. Although we don't know the $\delta$13C of peatland in this estuary, its range

probably overlays with the $\delta$13C of DOC at this estuary (-20 $\sim$ 30‰, so it cannot be identified from $\delta$13C signals. The detailed compositions of OM provided a powerful tool to make clear the mechanisms of OM at this estuary. Since the fieldwork was carried out at dry season, the input from peatland probably was minimal, particularly the particle input. Hence the compositions of POC didn't show a visible contribution from peatland. I agree with the authors that in the wet season the signals from peatland probably will increase.

---

## Short Comment (SC2) · 10 Oct 2019

I have some questions as following. Thank you in advance for your reply.

1. If the variation of D/L Glx ratios was related to DON? When you discussed the relationship between D/L Glx and salinity, did you only consider the carbon in Glx?

2. Why did you draw a dotted line in Figure 5a? I initially thought there was some linear relationship but finally found it was not.

3. In Figure 1b, S18, S17, S19, and S15 looked like in the downstream of Sibu, why did you mark them as freshwater?

<a href="#">Printer-friendly version</a>

<a href="#">Discussion paper</a>

---

## Short Comment (SC3) · 13 Oct 2019

Thank you for your interest in the manuscript. Here is the reply to your questions. 1. If the variation of D/L Glx ratios was related to DON? When you discussed the relationship between D/L Glx and salinity, did you only consider the carbon in Glx? First of all, we checked the D/L Glx and DON. Statistically, D/L Glx was not related to DON in the brackish estuary part at all (p=0.31), nor in the river part (fresh water only; p =-0.428). About the second question, amino acids are usually chiral (excluding a few like glycine) compounds (i.e., with D and L forms). Though abiotic racemization produces D form of amino acid from its corresponding L form, but in contemporary

aquatic systems, the key source for D form of amino acid is microbe and their activity (early diagenesis of OM). For example, bacteria produce and utilize certain D form amino acids to construct their cell membrane and hence some D-amino acids are key compounds that being found in peptidoglycan (Schleifer and Kandler, 1972). Also, when microbes utilized OM (they are usually heterotrophic), the D form amino acids usually accumulated (and hence D/L ratio increased). This is not only because of bacteria presence, but also largely due to the detritus of bacteria. As a total result, the D/L ratio is one of the proxies that can indicate the early diagenesis statues of OM (Davis et al., 2009) with higher D/L ratio usually means advanced degradation status of OM relative to lower D/L ratios within a given system. In the current manuscript, when we discuss the relation between D/L and salinity, we are trying to indicates the degradation status of OM along with salinity. We hope this answer is what you want.

2. Why did you draw a dotted line in Figure 5a? I initially thought there was some linear relationship but finally found it was not. Sorry for the misleading line. We meant to show the theoretical and conservative mixing trend along with salinity in this figure.

3. In Figure 1b, S18, S17, S19, and S15 looked like in the downstream of Sibu, why did you mark them as freshwater? Yes these stations are located down stream of Sibu, but when we were there doing the sampling and observation, indeed we observed salinity = 0 for all these stations. These stations showed a slightly higher conductivity (ranged between 36.9 to 124.4 $\mu$S/cm) when compared with Sibu station (for S14, its conductivity = 36.8 $\mu$S/cm), but when converted to salinity, it is still 0. In order to distinguish these zero-salinity station, we prepare figure 2-5 with x-axis including both conductivity and salinity unit.

---

## Short Comment (SC4) · 13 Oct 2019

Davis, J., Kaiser, K., and Benner, R.: Amino acid and amino sugar yields and compositions as indicators of dissolved organic matter diagenesis, Organic Geochemistry, 40, 343-352, 2009. Schleifer, K. H. and Kandler, O.: Peptidoglycan types of bacterial cell walls and their taxonomic implications, Bacteriological Reviews, 36, 407-477, 1972.

---

## Editor Comment (EC1) · Phillip Ford (Editor) · 12 Nov 2019

It has not been possible to find 2 reviewers to provide Biogeosciences with advice whether to accept your M/S into the Discussion Phase. We received one positive review some time ago, but a second reviewer has proved elusive despite multiple invitations. On the advice of the Chief Editor I have looked at your M/S and, in light of the reviewer's positive judgement, formed the view that it should be accepted into the Discussion phase subject to modification to address the following issues, primarily of Figure presentation, and minor typographical problems.

Figure 1a. Needs to have more geographical detail, place names, countries and island

names should to be included.

Figure 1b. The size of this Figure needs to be bigger so that the location of the sample sites can be discerned. The shading used does not clearly differentiate between the "potential area of Peat Swamp" and "mangroves". What is the significance of the dotted line? Does it represent the land sea boundary (high-tide mark). Its significance needs to be explained. The symbols for the estuary and freshwater stations are indistinguishable also. They need to be made larger and more distinctive.

Figure 2a-c. The symbols need to be larger and clearer. It is very hard to discriminate between the observations from the Rajang, Igan, and marine sites. Also, how can some points appear to "come and go"? In Figure 2 b there is a sample point on the Conductivity side at approx. 64 uS/cm but it's missing in Figures 2a and 2c. Similarly on the Salinity side of the figure: In Figures 2 a and 2c there is a single point on the salinity axis in the range 0 to 10, yet in Fig. 2b there are 4 points? The plotting of the freshwater on a much larger scale (Conductivity) axis than the estuarine samples (Salinity axis) seems to me to give undue weight to the minor differences between all the freshwater samples. Perhaps they should be averaged and shown with standard deviation, as the average freshwater end member on the Salinity axis. Do the minor differences in Conductivity have any spatial pattern along the Rajang River?

The captions to Figures 4 and 5 should explain that the dashed line is the conservative mixing line. The text needs to explain why only S1 was used in constructing the mixing line when potentially all the marine sites (S1, S22, S23, and S33) could have been used. Using the average of all these marine sites as the marine end member, and the average of the 8 freshwater sites in the Rajang River as the freshwater end member would, in my opinion, provide a more defensible mixing line as well as giving standard deviations of the end members, and thus an indication of uncertainties in the line location. The second sentence of the caption to Figure 5 is unclear ( "were" instead of "where"?) and needs to revised.
The text needs to be carefully read and corrected for minor mis-spellings and poor grammatical construction. See lines 163, 214, 223, 224,206, 246, 254/5, 284, 299, 313/4.

---

## Short Comment (SC5) · 3 Dec 2019

Figure 1: We revised figure 1 (both a and b) accordingly this time.

Figure 2: We revised figure 2a-c accordingly this time (symbols larger, clearer). About the "come and go" dots in fig. 2, this is because a few samples were missing. This includes the POC and TSM samples at station S16 (conductivity = $\sim$64 uS/cm), at station S4 (salinity = 4.8), station S25 (salinity = 11.7) and at station S29 (salinity = 4.3). We would mention this sample missing status in the materials and methods sections in the revised manuscript. About the suggestion for average all the fresh samples and show a mean value (instead of previous showing detailed conductivity in fig 2), we think

[Figure]

it makes sense and hence figure 2 are revised accordingly (namely only one point now stands for fresh endmember with stdev).

Figure 4 and 5: Yes the dashed line is the conservative mixing line and it will be added into the captions for fig.4 and fig. 5. For the linear mixing line in fig. 4 and fig.5, we think editor's suggestion for freshwater endmember is right and hence we revised the original fig. 4 and fig.5 accordingly. About the marine endmember, we think it is better to choose all the samples with salinity > 30 and take the average values of these marine sites as the marine endmember. Then a simple linear line connects both fresh and marine endmember should be the conservative mixing line. We attached the revised fig.4 and 5 in the follow and it will be revised in the final manuscript accordingly. Finally, we are happy to add some additional data from a few samples (which were just available from colleagues recently) to figure 4 and 5. So now the total data dots in revised figure 4 and 5 are slightly more than previous submitted figure 4 and 5. But due to similar sampling missing reason as figure 2, the particulate and dissolved dots do not exactly match to each other. But overall we think this flaw does not interfere the overall data pattern. The figure caption for fig.4 and fig. 5 are also revised and updated with more detail and explanation.

About the text mis-spellings and poor grammatical constructions, these sentences (lines 163, 214, 223, 224,206, 246, 254/5, 284, 299, 313/4.) will be revised the spelling/grammar problems in the revised manuscript. Thank you.

Following are the revised figures 1, 2, 4 and 5, as well as its figure captions.  

Figure 1. Study area and sampling stations. a) Location of Sarawak, Malaysia; and b) the Rajang with its estuary/river mouth background shown. Samples upstream of Sibu showed 0 salinity while downstream of Sibu showed salinity >0. Hence here from Sibu to Kapit is regarded as the fresh water section, and downstream of Sibu is regarded as the estuarine section.  

Figure 2. Distribution pattern of (a)TSM, (b) DOC and (c) POC along with conductivity/salinity in the Rajang. The location of salinity = 0 is at Sibu (Fig. 1b).  

Figure 4. GABA% distribution pattern from fresh water to estuary in the Rajang: a) dissolved and b) particulate. The dashed line indicates the linear mixing line, with the fresh and marine endmember calculated as the means of all fresh samples (S = 0) and all offshore samples with salinity >30, respectively. The calculated fresh and marine endmembers are also shown in both plots (as brown triangle and purple diamond) and note that these two dots are not from real field samples. The error bar indicates the standard deviation of both endmembers, respectively.  

Figure 5. Same as figure 4, but for D/L-Glx.
* * *
[Figure]

[Figure]

Figure 1. Study area and sampling stations. a) Location of Sarawak, Malaysia; and b) the Rajang with its estuary/river mouth background shown. Samples upstream of Sibu showed 0 salinity while downstream of Sibu showed salinity >0. Hence here from Sibu to Kapit is regarded as the fresh water section, and downstream of Sibu is regarded as the estuarine section.

**Fig. 1.** figure 1-revised

[Figure]

Figure 2. Distribution pattern of (a)TSM, (b) DOC and (c) POC along with conductivity/salinity in the Rajang. The location of salinity = 0 is at Sibu (Fig. 1b).

**Fig. 2.** figure 2-revised

[Figure]

Figure 4. GABA% distribution pattern from fresh water to estuary in the Rajang: a) dissolved and
b) particulate. The dashed line indicates the linear mixing line, with the fresh and marine endmember
calculated as the means of all fresh samples (S = 0) and all offshore samples with salinity >30,
respectively. The calculated fresh and marine endmembers are also shown in both plots (as brown
triangle and purple diamond) and note that these two dots are not from real field samples. The error
bar indicates the standard deviation of both endmembers, respectively.

**Fig. 3.** figure 4-revised

[Figure]

Figure 5. Same as figure 4, but for D/L-Glx.

**Fig. 4.** figure 5-revised

---

## Editor Comment (EC2) · Phillip Ford (Editor) · 5 Dec 2019

Associate editor's comments: bg-2019-157.The non-conservative distribution pattern of organic matter in Rajang, a tropical river with peat land in its estuary. Lead author: Zhuyin Zhu.

It has not been possible to find 2 reviewers to provide Biogeosciences with advice whether to accept your M/S into the Discussion Phase. We received one positive review some time ago, but a second reviewer has proved elusive despite multiple invitations. On the advice of the Chief Editor I have looked at your M/S and formed the view that it should be accepted into the Discussion phase subject to modification to

address the following issues, primarily of Figure presentation, and minor typographical problems. Figure 1a. Needs to have more geographical detail, place names, countries and island names should to be included. Figure 1b. The size of this Figure needs to be bigger so that the location of the sample sites can be discerned. The shading used does not clearly differentiate between the "potential area of Peat Swamp" and "mangroves". What is the significance of the dotted line? Does it represent the land sea boundary (high-tide mark). Its significance needs to be explained. The symbols for the estuary and freshwater stations are indistinguishable also. They need to be made larger and more distinctive. Figure 2a-c. The symbols need to be larger and clearer. It is very hard to discriminate between the observations from the Rajang, Igan, and marine sites. Also, how can some points appear to "come and go"? In Figure 2 b there is a sample point at Conductivity side at approx. 64 uS/cm but it's missing in Figures 2a and 2c. Similarly on the Salinity side of the figure: In Figures 2 a and 2c there is a single point in the salinity in the range 0 to 10, yet in Fig. 2b there are 4 points? The plotting of the freshwater on a much larger scale (Conductivity) axis than the estuarine samples (Salinity axis) seems to me to give undue weight to the minor differences between all the freshwater samples. Perhaps they should be averaged and shown with standard deviation, as the average freshwater end member on the Salinity axis. Do the minor differences in Conductivity have any spatial pattern along the Rajang River? The captions to Figures 4 and 5 should explain that the dashed line is the conservative mixing line. The text needs to explain why only S1 was used in constructing the mixing line when potentially all the marine sites (S1, S22, S23, and S33) could have been used. Using the average of all these marine sites as the marine end member, and the average of the 8 freshwater sites in the Rajang River as the freshwater end member would, in my opinion, provide a more defensible mixing line as well as giving standard deviations of the end members, and thus an indication of uncertainties in the line location. The second sentence of the caption to Figure 5 is unclear ( "were" instead of "where"?) and needs to revised. The text needs to be carefully read and corrected for minor mis-spellings and poor grammatical construction. See lines 163, 214, 223,

224,206, 246, 254/5, 284, 299, 313/4.

Phillip Ford 12 November 2019

---

## Author Comment (AC2) · 6 Dec 2019

Figure captions.

Figure 1. Study area and sampling stations. a) Location of Sarawak, Malaysia; and b) the Rajang with its estuary/river mouth background shown. Samples upstream of S5 showed 0 salinity while downstream of S5 showed salinity >0. Hence here from S5 to S10 is regarded as the fresh water section, and downstream of S5 is regarded as the estuarine section.

Figure 2. Distribution pattern of (a)TSM, (b) DOC and (c) POC along with conductivity/salinity in the Rajang. The fresh water dot stands for all samples with S = 0 and the error bar corresponds to the standard deviation. The marine dot is S1.

Figure 4. GABA% distribution pattern from fresh water to estuary in the Rajang: a) dissolved and b) particulate. The dashed line indicates the linear mixing line between fresh and marine endmembers. The fresh water endmember (brown triangle), it is calculated as the means of all fresh samples (S = 0), and the marine endmember (purple diamond) is calculated as the means of all offshore samples with salinity >30. The error bar indicates the standard deviation.

Figure 5. Same as figure 4, but for D/L-Glx.

[Figure]

Figure 1.

[Figure]

Figure 2.

[Figure]

Figure 4.

[Figure]

Figure 5.

---

## Author Response (AR1)

Dear editor, We thank editor for the help in improving the original manuscript. Now the manuscript is revised. The revised points includes:

1 changes to figure 1a: we add more details in geophysical names, and also a revise in figure 1b with updating the station symbols, and a better way in displaying the mangrove and peatland in the estuary region

2 changes to figure 2 (a,b,c): we merged all fresh water sample dots together into one dot (but with stdev).

3 explain the 'come and go' dots: in section 2.2 we added the missing sample list (line 152-156), and also at the end of section 2.3 (line 186/7) we state the missing sample situation.

4 revising fig 4 and 5: the linear mixing line is revised, with fresh and marine endmember recalculated, in the way as we previously replied to editor in the interactive discussion forum.

5 the minor spelling/grammar problem (lines 163, 214, 223, 224, 206, 246, 254/5, 284, 299, 313/4 in the origin manuscript) was revised this time

6 the spelling of authors is also revised: Zhuoyi now changed to Zhuo-Yi; Youyou now changed to You-You. The affiliation of Zhuoyi Zhu is also revised ('school of oceanography' is now deleted).

7 We also revised the last paragraph in section 4.1 (line 300-304), as we mentioned/response during the interactive discussion forum. Now this paragraph reads:

Although particulate OM had a lower D/L ratio than dissolved OM (Fig. 6), it should be noted

that this does not mean dissolved OM is more aged or degraded than particulate OM. Riverine POM

and DOM usually show different ages (Bianchi and Bauer, 2011), while selective

desorption/adsorption of bacteria and related detritus between particulate and dissolved phase also

strongly modifies the biomarker-indicated degradation status of OM (Dittmar et al., 2001a).

Following is a marked-up file for the manuscript. In this marked-up file, all the revision are tracked and shown.

Thank you Zhuoyi

**1 The non-conservative distribution pattern of organic matter in**

2

**۷**

**Rajang, a tropical river with peatland in its estuary**

ZhuoyiZhuo-Yi Zhu1\*, Joanne Oakes2, Bradley Eyre2, YouyouYou-You Hao1, Edwin Sien Aun
 Sia3, Shan Jiang1, Moritz Müller3, Jing Zhang1

[revised manuscript text omitted]

---

## Author Response (AR2)

Dear Phillip,

With the help from one of the native-English-speaker co-authors, the manuscript is now language-edited again thoroughly, including all the minor grammatical issues you've mentioned in a previous round.

About the figures-appeared-twice problem, I checked the uploaded file. The figures now appear only once. I have no idea why previously we encountered an appearing twice problem. In this version, if you still find such a problem in the uploaded file, or there is anything I should do, please let me know.

Thank you for your suggestion and help.

Zhuoyi

[revised manuscript text omitted]

Figure 1.

[Figure]

Figure 2.

[Figure]

[Figure]

Figure 3.

[Figure]

Figure 4.

[Figure]

Figure 5.

[Figure]

Figure 6.

[Figure]

Figure 7.

[Figure]

Figure 8.